# Combined Effects of Temperature and Dietary Lipid Level on Body Composition, Growth, and Freshness Profile in European Seabass, *Dicentrarchus labrax*

**DOI:** 10.3390/ani13061068

**Published:** 2023-03-15

**Authors:** Patrícia G. Cardoso, Odete Gonçalves, Thais Cavalheri, Vânia E. Amorim, Weiwei Cao, Diogo A. M. Alexandrino, Zhongjun Jia, Maria F. Carvalho, Paulo Vaz-Pires, Rodrigo O. A. Ozório

**Affiliations:** 1CIIMAR—Interdisciplinar Centre of Marine and Environmental Research, Terminal de Cruzeiros de Leixões, Av. General Norton de Matos, S/N, 4450-208 Matosinhos, Portugal; 2State Key Laboratory of Soil and Sustainable Agriculture, Institute of Soil Science, Chinese Academy of Sciences, Nanjing 210008, China; 3Department of Environmental Health, School of Health, Polytechnic of Porto, Rua Dr. António Bernardino de Almeida 400, 4200-072 Porto, Portugal; 4ICBAS—School of Medicine and Biomedical Sciences, University of Porto, R. Jorge Viterbo Ferreira, 228, 4050-313 Porto, Portugal

**Keywords:** climate change, fish quality, feeding, fitness, freshness, fish-spoiling bacteria

## Abstract

**Simple Summary:**

The effects of increasing temperature and dietary lipid level on the body composition, growth performance, and freshness profile of the European seabass (*Dicentrarchus labrax*) were evaluated through a fish trial lasting 56 days. Findings demonstrated that fish reared at 24 °C presented a lower lipid level and a higher daily growth index than those reared at 20 °C. On the other hand, the sea bass condition index did not change among treatments. Additionally, sensory analysis (the Quality Index Method) and microbiological analysis revealed that fish reared at 24 °C showed better freshness conditions than those at 20 °C. Nevertheless, the dietary lipid level did not have any influence on fish freshness conditions. Therefore, our data suggest that the increase in temperature to 24 °C is beneficial for the growth and freshness profile of this particular species in aquaculture.

**Abstract:**

A fish trial was carried out to evaluate the combined effects of temperature and dietary lipid level on the body composition, growth performance, and freshness profile of the European seabass (*Dicentrarchus labrax*). Fish were kept for 56 days at 20 °C and 24 °C and fed on two diets, with 16% and 20% lipid. At the end of the trial, fish were euthanized at two temperature conditions (0.6 °C or −0.6 °C) and kept on ice for 10 days at 4 °C to evaluate their freshness condition. Findings demonstrated that fish reared at 24 °C presented a lower lipid level and a higher daily growth index than those at 20 °C. Additionally, sensory analysis (Quality Index Method—QIM) and microbiological analysis revealed that fish reared at 24 °C showed better freshness conditions than those at 20 °C. However, the 16S rRNA metabarcoding analyses revealed a higher proliferation of genera associated with fish-spoiling bacteria in the skin microbiome of fish reared at 24 °C, i.e., Vibrio and Acinetobacter, which was not observed in the skin microbiome of fish reared at 20 °C. Nevertheless, the dietary lipid level did not have any influence on fish freshness. Therefore, our data suggest that the increase in temperature to 24 °C is beneficial for the growth and freshness profile (lower QIM and lower CFUs/cm^2^) of this particular species. Additionally, the lower euthanasia temperature (−0.6 °C) seems to lead to higher fish freshness than the normal temperature (0.6 °C).

## 1. Introduction

Coastal systems are extremely important to the planet due to the goods and services they provide. However, they have been threatened by numerous stressors, namely global climate change. During the last decades, an increase in the sea surface temperature has been observed, and it is expected to increase by 1.1–6.4 °C until the end of the century [1]. Temperature is one of the most relevant environmental drivers controlling the functioning of aquatic communities [2,3,4]. Fish have the ability to balance their internal temperature with the external environment, being particularly affected by environmental modifications. [2]. For example, fish growth can be directly affected by temperature through modifications in feed consumption and nutrient requirements [5]. Additionally, culture temperature may also affect the body’s fat deposition as well as its fatty acid profile. These variations could affect the nutrient demands of the fish as well as their organoleptic characteristics when raised at different temperatures. Moreover, according to available data, high dietary lipid levels may trigger an increase in lipid deposition in fish and alter the quality of the flesh in terms of freshness, storage stability, processing yields, and organoleptic and physical properties [6]. So, a combination of both factors (i.e., increased temperature and high dietary lipid levels) can influence fish freshness. Freshness is a fish quality attribute and probably the most important quality criterion for most fish products [7]. The loss of freshness is a very rapid and complex process that depends on the interaction of multiple reactions (i.e., physical, chemical, and microbiological) [8]. In order to evaluate fish freshness, several methods can be used, with sensory analysis being one of the most commonly used, and in particular, the quality index method (QIM). It is based on the evaluation of changes in relevant characteristics of seafood, such as skin, eyes, gills, and odor, among others, and can be defined using demerit points from 0 to 3 (less fresh) [9]. An older sensory method for fish quality assessment is the UE Freshness Grading (or EC Scheme), which includes 3 levels: E (Extra, the highest quality), A (good quality), and B (satisfactory quality). Below level B (sometimes called unfit or C), fish is not acceptable for human consumption, thus it is discarded or rejected [10]. Other complementary methods to the sensory analysis (i.e., physical and microbiological analyses) can be used for determination of the freshness/quality of fish products [11]. The physical analysis is based on the principle of measurement of muscle dielectric properties through the use of specific equipment, such as the Torrymeter [12]. As freshness declines with time, it is expected that the electric resistance of cell membranes in fish muscle tissue will decrease, so electric impulses will cross fish tissues more easily with time [11,13]. The presence and activity of the microbiological community in seafood influence spoilage and fish storage time, so its evaluation is crucial to evaluating fish freshness conditions.

The European seabass, *Dicentrarchus labrax*, is a coastal marine teleost that lives in shallow waters (<100 m) with a wide geographical distribution through the coasts of the Mediterranean Sea, the Black Sea, and the eastern Atlantic Ocean from Norway to Morocco, the Canary Islands, and Senegal [14]. It is also extensively farmed under different conditions (e.g., open and semi-open culture systems), so it is able to cope with daily and seasonal oscillations in environmental conditions, namely sea surface temperature [15]. It can reach a maximum length of 1 m, but rarely exceeds 50 cm [16]. The European seabass is carnivorous, feeding on plankton, fish, and crustaceans [17]. The optimal culture temperature of D. labrax ranges between 20 and 24 °C [18], reaching sporadically higher temperatures in the ponds.

In the literature, there are many studies that have demonstrated the impacts of increased temperature on the growth and physiology of aquaculture species [19,20,21], but there are no records about their impact on fish freshness profiles. So, the novelty of the present study is to breakthrough the state of the art and evaluate the combined effects of increased temperature and diets with different lipid levels on the composition, growth, and freshness profiles of the seabass, *Dicentrarchus labrax*, and infer about the best conditions in order to obtain the freshest specimens.

## 2. Materials and Methods

According to the guidelines on the protection of animals used for scientific purposes from the European Directive 2010/63/UE. The present work was performed under the supervision of an accredited expert in laboratory animal science by the Portuguese Veterinary Authority (1005/92, DGV-Portugal, following FELASA category C recommendations).

### 2.1. Fish and Rearing Conditions

Before the trial, fish were kept in a maintenance system for 6 months in tanks of 250 L and fed a commercial diet (47% protein and 15% lipids).

Two hundred and four juvenile seabasses (16.8 ± 0.6 cm; 59.6 ± 3.6 g) were randomly distributed in 12 fiber-glass tanks of 200 L each (n = 3 tanks/treatment; 17 individuals/tank), in a factorial design manipulating temperature [ambient temperature (20 °C) (average summer sea surface temperature for the Portuguese coast) and warming (24 °C—the future sea surface temperature warming scenario in 2100 (+4 °C) [22] and two diets (16% and 20% lipid) for 56 days (Figure 1), in tanks with a mechanical filter, biofilter, skimmer, UV filter, and aeration system, and were maintained in a constant photoperiod (14 L: 10 D). Fish were fed twice a day until satiety. Tanks were monitored daily for O_2_ (7.5 ± 0.3 mg L^−1^), pH (7.2 ± 0.17), salinity (35.5 ± 0.7 ppt), ammonia (0.4 ± 0.19 mg/L), and nitrites (0.79 ± 0.41 mg/L). The temperature was kept according to the experimental design at 20.2 ± 0.19 °C and 23.9 ± 0.03 °C.

### 2.2. Experimental Diets

Diet 1 (D-4 Alterna 2P) was formulated to contain 46% protein and 16% lipid, whereas Diet 2 (L-4 Alterna 2P) contained 46% protein and 20% lipid. Both diets were formulated and commercialized by Skretting Spain SA (see in detail Appendix A).

### 2.3. Fish Sampling and Storage

After the trial, fish were euthanized using a water-ice mixture of 1:1 (WI) or 1:2 (WI) as the killing method to obtain 2 different temperature conditions: 0.6 °C (1:1) and −0.6 °C (1:2), following a similar procedure [23]. These two temperatures were controlled with a thermometer, and whenever necessary, more ice was added to the water. At each condition, 12 fish from each treatment (4 fish/tank) were selected and immediately stored in ice at 0–4 °C for 10 days in self-draining boxes, and ice was added as needed along the storage. At day 0 (D0, immediately after death) and day 10 (D10), fish were collected from the boxes for the freshness evaluation. Additional fish were used for body composition analyses. 

### 2.4. Composition Analysis

Fish previously frozen (n = 3 fish/tank) were pooled, cut into pieces without thawing, and ground homogeneously with a meat mincer. Afterwards, the homogenized sample was freeze-dried and stored at −20 °C. An aliquot of each homogenized sample was analyzed for dry matter (AOAC 2003.05; 105 °C; 24 h); lipid content (crude lipid by extraction with petroleum ether (Soxtherm; Gerhardt, Germany)); crude protein (N × 6.25; N = nitrogen) using a Leco nitrogen analyzer (Model FP528; Leco Corporation; St. Joseph, MI, USA); and gross energy (adiabatic bomb calorimeter; Model Werke C2000; IKA; Staufen, Germany).

### 2.5. Growth Performance 

At each tank, 8–15 fish were sampled for determination of initial body weight (IBW), final body weight (FBW), final length (L), and feed intake (FI) during the exposure period.

Feed conversion ratio (FCR) was calculated based on the following equation [24]:FCR = FI/BW,
where FI corresponds to the feed intake and BW is the body weight gain.

The condition index was calculated according to [24] as follows:CI = FBW/L^3^ × 100

Also, the daily growth index (DGI) was calculated as follows:DGI = 100 × (FBW^1/3^ − IBW^1/3^)/trial duration.

And the protein efficiency ratio (PER) was calculated as follows:PER = BW/crude protein intake, where: 
Crude protein intake = FI × % diet protein/100

### 2.6. Freshness Profile

#### 2.6.1. Sensorial Analysis 

Two different methods were used for sensory evaluation: (1) the Quality Index Method (QIM) [25] and (2) the European Scheme (EC) method [26]. Both were applied individually to all fish on each sampling day. According to the QIM scheme (from D0 to D10), quality attributes for appearance/texture, including eyes, gills, skin, mouth, and anal area, were analyzed along degradation time. Each attribute was evaluated according to descriptions, ranging from 0 (most fresh) to 3 (less fresh), and the final value corresponds to the sum of the scores of the different attributes [11,27]. Regarding the EC scheme, the quality attribute of this list was evaluated as “Extra”, “A”, “B”, or Rejected [27]. “Extra” animals ranged from 3–2.7; those from category "A" ranged from 2.7–2; and the ones from category “B” are between 2–1, according to Council Regulation nº 103/76.

#### 2.6.2. Physical Analysis

To assess the muscle dielectric properties, a torrymeter (TRM 295, Distell, West Lothian, Scotland, UK) was used according to the equipment user manual.

The same anterior-dorsal area for each fish was selected for measurements. For each fish, 8 readings (4 on each side) were carried out. Both sides of the fish were analyzed after removing the residual ice. The electrodes were cleaned between measurements, protected with cling film, and placed on ice to maintain the temperature around 0 °C. This procedure allowed to keep the electrodes at a similar temperature to that of the fish skin, avoiding some equipment damages.

#### 2.6.3. Microbiological Analysis

For the microbiological analysis, it was decided to analyze just the skin tissues since, according to the literature, the number of bacterial counts on the skin and gills usually yield proportional results [28,29]. Fresh fish usually present skin bacterial counts between 10^2^–10^3^, while the gills present very similar numbers (between 10^2^–10^4^), and it is known that these numbers increase proportionately as freshness is lost. In addition, we consider that the skin is more relevant to the consumer since, in some cases, the gills are removed before the fish reaches the consumer. So, in terms of cost-benefit, the study of the microbiology of the skin is more beneficial. Therefore, sampling of skin mucus was performed by swabbing the skin surface (3.14 cm^2^) with sterile cotton swabs at D0 and D10, where two swabs were collected on each side of the fish (n = 12 fish/treatment). Microorganisms attached to the swabs were released by immersing the swabs in tubes containing 2 mL of 0.9% saline solution, followed by a vigorous vortexing of these tubes. Serial decimal dilutions were performed and used to inoculate, in duplicate, iron agar Lyngby plates by the drop method (20 µL). Plates were incubated at 20 °C for 48 h, after which total viable bacteria (TVC) and H_2_S-producing bacteria were counted. Results of colony counts were converted to the logarithm of cfu/cm^2^.

### 2.7. Taxonomic Profiling of the Skin Microbiome by 16S rRNA Gene Metabarcoding Sequencing

The mucus of the skin was swabbed from fish sacrificed at 0.6 °C following a similar procedure as described in Section 2.6.3, and the swabs were stored in sterile Falcon tubes at −80 °C until further processing. DNA was isolated from the swabs using the PureLink™ Microbiome DNA Purification Kit (ThermoFisher Scientific, Waltham, MA, USA) according to the instructions of the manufacturer. DNA quality and concentration were controlled in a Denovix DS-11 spectrophotometer/fluorometer. Samples showing suitable DNA concentrations and quality ratios were submitted to high-throughput sequencing of the V4-V5 hypervariable region of the 16S rRNA gene on the Illumina MiSeq platform, as described in [30]. Briefly, 16S rRNA amplicons were amplified by PCR using the bacterial primer pair 515F/907R, further purified by a 1.2% agarose gel electrophoresis, and mixed at equimolar concentrations for sequencing. Sequence library preparation was done with the TruSeq Nano DNA LT Sample Prep Kit Set A, and the sequencing was performed with the MiSeq Reagent Kit v3 (600 cycles). All sequence data used in this study were placed in the European Nucleotide Archive with the accession number PRJEB56124. 

### 2.8. Data Analyses

For the composition analysis, growth evaluation, and condition index, 2-way ANOVAs (factors: temperature and diet) were applied to assess statistical differences among treatments.

A 3-way ANOVA on ranks (factors: temperature, diet, and days) was applied to test for statistical differences in sensorial, physical, and microbiological analyses among treatments.

Prior to the statistical analyses, the Kolmogorov–Smirnov test and the Levene’s test were used to check for normality and homogeneity of variances in the raw data, respectively. [31]. These analyses were done using Statistica 7 software.

For the generated 16S rRNA amplicon sequencing datasets, bioinformatics analyses were performed using the QIIME pipeline [32] and the phyloseq package in the R environment (v.3.6.1) [33], as described elsewhere [30]. To track the emergence of potential fish-spoiling bacteria during the 10-day ice storage period, the abundances (as read counts) of bacterial genera mostly known to accommodate such microorganisms, namely *Pseudomonas*, *Acinetobacter*, *Vibrio*, *Shewanella*, and *Alteromonas* [34,35,36,37], were log_10_-transformed for a direct comparison of their prevalence and distribution between D0 and D10.

## 3. Results

### 3.1. Compositional Analysis

Lipid levels were significantly higher in fish reared at 20 °C (41–44%) than those at 24 °C (39–41%) (2-way ANOVA, F_(1, 8)_ = 12.21, *p* < 0.05). Also, fish fed with 20% lipid presented significantly higher body lipid content (2-way ANOVA, F_(1, 8)_ = 6.25, *p* < 0.05) than those fed with 16% lipid (Figure 2A).

On the other hand, protein content was significantly higher in fish reared at 24 °C (2-way ANOVA, F_(1, 20)_ = 9.81, *p* < 0.05) and fed with a lower dietary lipid level (2-way ANOVA, F_(1, 20)_ = 8.20, *p* < 0.05) (Figure 2B).

Regarding energy, a significant interaction between factors (temperature and diet) was observed (2-way ANOVA, F_(1, 20)_ = 4.69, *p* < 0.05). Apparently, fish reared at 20 °C and fed 16% lipid had significantly lower body energy content than the other treatments (Figure 2C).

### 3.2. Growth Evaluation and Condition Index

Growth performance did not vary among the treatment groups, except for the daily growth index (DGI). The DGI varied between temperatures, being higher at 24 °C than at 20 °C (2-way ANOVA, F_(1, 8)_ = 11.3, *p* < 0.05) (Table 1). Despite the fact that there were no significant differences among treatments, body weight gain (BW) showed a tendency to increase in fish reared at 24 °C when compared to fish reared at 20 °C. For the remaining parameters, there were no significant differences among treatments.

### 3.3. Sensorial Analysis

#### 3.3.1. Quality Index Method (QIM)

Regarding QIM analysis, both fish groups sacrificed at 0.6 °C and −0.6 °C increased their demerit points (meaning lower freshness) from D0 to D10, reaching higher values when sacrificed at 0.6 °C (10.8–11.8) than at −0.6 °C (8.2–8.9) (Figure 3A,B). In addition, a significant interaction between temperature and storage days was observed for fish sacrificed at 0.6 °C (3-way ANOVA on ranks, F_(1, 88)_ = 6.35, *p* < 0.05), and on D10, fish reared at 20 °C showed significantly higher QIM values than those at 24 °C. This was only visible for the ones sacrificed at 0.6 °C. Concerning the increment of demerit points between D0 and D10 and comparing both euthanasia methods, it is clear that the fish sacrificed at a lower water temperature reached significantly lower QIM values, indicating higher freshness (1-way ANOVA, F = 108.89, *p* < 0.05) (Figure 3C).

#### 3.3.2. EC Scheme

Regarding the EC Scheme, there was a significant decline in demerit points from D0 to D10 in both euthanasia methods (at 0.6 °C: 3-Way ANOVA on ranks, F_(1, 88)_ = 2926.13, *p* < 0.05; at −0.6 °C: 3-Way ANOVA on ranks, F_(1, 88)_ = 613.35, *p* < 0.05) (Figure 4A,B), which means a decline of the freshness profile. No significant differences between temperatures and diets were observed. In this case, no significant differences between the two types of euthanasia methods were observed (*p* > 0.05) (Figure 4C).

#### 3.3.3. Physical Analysis (Torrymeter—TRM)

The TRM analysis was quite similar to the EC scheme results. In fish sacrificed at 0.6 °C, a significant interaction among the three factors was observed (3-way ANOVA on ranks, F_(1, 88)_ = 7.78, *p* < 0.05), and a significant decline in TRM values from D0 to D10 was observed. For those sacrificed at a lower temperature, there was a significant decline in TRM values between sampling times (3-way ANOVA on ranks, F_(1, 88)_ = 448.72, *p* < 0.05); however, no significant differences between euthanasia methods were observed (*p* > 0.05) (Figure 5C).

#### 3.3.4. Microbiological Analysis

For fish sacrificed at 0.6 °C, the total viable counts (TVC) recorded in the skin increased significantly from D0 to D10, and in the latter, fish reared at 20 °C presented significantly higher TVC than those at 24 °C (3-way ANOVA on ranks, F_(1, 88)_ = 5.27, *p* < 0.05) (Figure 6A). For fish sacrificed at −0.6 °C, there was also an increase in TVC from D0 to D10, but in the latter, TVC at 20 °C was lower than at 24 °C (3-way ANOVA on ranks, F_(1, 88)_ = 12.75, *p* < 0.05) (Figure 6B). Also, an interesting result was that fish sacrificed at −0.6 °C had a higher number of TVC at D0 (10^1^–10^2^ cfu/cm^2^) than those sacrificed at 0.6 °C (<10^1^ cfu/cm^2^).

Regarding the increment of TVC from D0 to D10, a significant interaction between death and temperature was observed (3-way ANOVA on ranks, F_(1, 78)_ = 35.45, *p* < 0.05). For fish reared at 20 °C, there were significant differences between the two euthanasia methods (i.e., lower temperature death corresponded to a lower TVC in the skin) (Figure 6C).

### 3.4. Taxonomic Profiling of the Skin Microbiome by 16S rRNA Gene Metabarcoding Sequencing 

Skin swabs of fish sacrificed at 0.6 °C, at D0 and D10, were selected for microbiome analysis through high-throughput sequencing of 16S rRNA amplicons. Taxonomic profiling at D0 revealed that the skin microbiome of all fish presented similar diversity and richness, regardless of rearing and dietary conditions (Figure 7A). Proteobacteria and Bacteroidetes phyla dominated the skin microbiome of *D. labrax*, though the latter phylum was found particularly abundant in fish reared at 20 °C and with a 16% lipid diet (Figure 7B). At the genus level, *Pseudomonas* was the dominant phylotype across all samples, in some cases accounting for over 90% of representation in the bacterial communities (Figure 7C). Some exceptions to this were noted in fish reared at 20 °C and fed 16% lipid, where an unusual prevalence of *Saprospira* (S15) and *Planctomyces* (S17) were observed in some samples (Figure 7C). In addition, *Tenacibaculum* bacteria were found to have an important representation alongside *Pseudomonas* in fish reared at 24 °C (Figure 7C). 

Ten days (D10) of ice storage had no clear impact on the α and β-diversity metrics among the different rearing and dietary conditions (Figure 8A). Proteobacteria remained the dominant phylum, but Bacteroidetes decreased their representation when compared with D0, being replaced, in some cases, by the Chlamydiae phylum (Figure 8B). *Pseudomonas* was, once again, the dominant phylotype across all experimental conditions at D10, followed by *Acinetobacter* and bacteria accommodated within the *Enterobacteriaceae* family, which were not among the 10 most abundant genera in the D0 samples (Figure 8C). Enterobacteria were particularly common in fish reared at 24 °C (Figure 8C). On the other hand, *Tenacibaculum* bacteria, which were prevalent at D0 in the skin microbiome of fish reared at 24 °C, did not show a relevant representation in the microbiome after the 10-day ice storage period.

By tracking the prevalence and distribution of bacterial genera associated with fish-spoiling bacteria, it was possible to detect some relevant shifts in the abundance of some of these bacterial genera (Figure 9). For instance, *Vibrio* and *Acinetobacter* bacteria significantly increased (*p* < 0.05) their abundance from D0 to D10 in the skin microbiome of fish reared at 24 °C and fed 16% lipid (Figure 9). For those fish reared at the same temperature and fed with 20% lipid, a similar trend was observed for these bacterial genera, but the lack of sample size between D0 (N = 4) and D10 (N = 6) precluded any statistical validation of this observation. Conversely, in fish reared at 20 °C, there was no apparent proliferation of *Pseudomonas*, *Acinetobacter*, *Vibrio*, *Shewanella*, or *Alteromonas* during ice storage, as the abundance of these taxa either remained similar or showed a significant decrease during the 10-day post-mortem period (Figure 9).

## 4. Discussion

Over the last years, several works have evidenced the effects of climate change at different levels, however, little is known about the impact of climate heating on the conditions of fish production in aquaculture and implications on their freshness profile.

Our findings demonstrated that higher resting temperatures led to a reduction in body lipid content, whereas body lipid content increased with the dietary lipid level, as expected. These results seem to be in accordance with [21,38], which concluded that higher temperatures resulted in lower lipid deposition. Regarding the effect of dietary lipid level, other studies also demonstrated that high dietary lipid could induce fat deposition [39].

On the other hand, the protein content declined in fish reared at 20 °C and fed a 20% lipid diet. Regarding the effect of temperature, our findings are in accordance with [40], for the Asian seabass, that observed that body protein was lower at lower temperatures. However, a contrary response was observed for the pikeperch, *Sander lucioperca* (i.e., higher protein content when exposed to a lower temperature) [41]. In other works, no effect of temperature on body protein content was observed [42]. In addition, concerning the effects of diet, our results are in agreement with those of [43], who observed a lower percentage of crude body protein in fish exposed to higher dietary lipids.

Energy content seems to be lower at lower temperatures and in diets with lower lipid content. Temperature has a crucial effect on the physiology of animals. As a result of global warming, increasing temperatures can affect fish populations by decreasing their performance when temperatures rise above an optimal range of values, which can lead to a greater consumption of energy [44]. In the present study, it seems that 24 °C is a more adequate temperature for sea bass, allowing it to store more energy than at 20 °C with a 16% lipid diet. On the other hand, the dietary lipid level had no effect on body energy storage.

In addition, our results also demonstrated that the increase in temperature from 20 °C to 24 °C had a positive effect on growth performance. Fish reared at 24 °C had a significantly higher daily growth index than those exposed to 20 °C. This pattern was previously observed in other studies, like the one in [45], in which a specific growth rate increased with temperature up to a maximum at 25 °C. According to the literature, growth rate increases with increasing temperature; however, at extreme temperatures, the effect can be negative [35].

Concerning fish freshness and the sensorial analysis, it was observed that QIM values, after 10-day ice storage, were significantly lower at 24 °C and at 0.6 °C (euthanasia temperature). This is in accordance with the lower lipid content, which corresponds to a higher freshness level. The freshness level increased (QIM decreased) under the euthanasia conditions at a lower temperature. So, at 0.6 °C, fish showed a higher increment (» 2 demerit points) of QIM from D0 to D10, compared with the −0.6 °C euthanasia condition. In the literature, there is a great lack of knowledge about the possible effects of increased temperature due to climate heating on the freshness profile of fish species. A recent paper from [46] has demonstrated that there is not a typical seasonal freshness profile. This will vary according to the fish species. The same result is visible here. According to the EC scheme, there were no significant effects of temperature or diet on the freshness profile of the seabass. On the other hand, we observed a clear temporal degradation of the quality of fish during the ice storage period, as already demonstrated previously [46]. The physical analyses of fish sacrificed at 0.6 °C confirmed that fish reared at 24 °C showed higher freshness than those exposed to 20 °C. This difference was not visible for fish sacrificed at −0.6 °C.

The screening of fish-spoilage activity and of the commonly associated bacterial taxa implemented in this study corroborated the temporal degradation of fish quality during the period of ice storage. Fish reared at 24 °C presented a lower number of TVC compared to those at 20 °C (sacrificed at 0.6 °C), which is in accordance with the previous parameters of freshness evaluation (i.e., sensory and physical analyses). Regarding the taxonomic profiling of fish skin microbiome by 16S rRNA gene metabarcoding, this analysis allowed to elucidate, through culture-independent methods, how bacterial taxa in general and spoilage-associated taxa evolved over the time of storage of fish on ice. Therefore, our findings demonstrated that a higher proliferation of genera associated with fish-spoiling bacteria occurred in the skin microbiome of fish reared at 24 °C, i.e., *Vibrio* and *Acinetobacter*, than was observed in the skin microbiome of fish reared at 20 °C and euthanized at 0.6 °C. Despite the observed increased proliferation of potentially degradative bacteria in this subset of specimens, the impact on the organoleptic features of European seabass was minimal within the 10-day storage on ice, suggesting that longer storage periods would be needed to see a significant decrease in the freshness profiles as a consequence of the continuous growth of these bacterial taxa. It is also possible that the decreased freshness profile in fish reared at 20 °C could have been influenced by other non-microbial factors, such as enzymatic action and lipid oxidation [8].

It is also curious that the number of TVC at D0 was three times higher in fish sacrificed at −0.6 °C compared with fish sacrificed at 0.6 °C. This can be related to the fact that fish sacrificed at −0.6 °C produced more mucus than at 0.6 °C (visual observations), which can promote the development of a bacterial community. This mucus production can be related to stressful environmental conditions (e.g., salinity, oxygen, nutrients, and temperature changes) for the fish; in this case, the decline in temperature. According to previous studies, changes in skin mucus production and composition are responses of the fish to cope with changes in the surrounding water [47]. Effectively, many authors suggested that the number of skin mucous cells could be used as a stress index for fish [48].

## 5. Conclusions

This study allowed us to conclude that 24 °C seems to be ideal for the rearing of European seabass when compared to 20 °C, allowing for lower lipid content, higher growth rates, and a better freshness profile. Nevertheless, fish reared at 24 °C showed higher growth of spoiling bacteria, though this was not reflected in the freshness profile of the species. The lipid level did not have any significant effect on the studied parameters of the European seabass. Moreover, the euthanasia method at a lower temperature (−0.6 °C) led to a higher freshness profile, despite the fact that at T0 the TVC was higher than at a 0.6 °C killing temperature.

## Figures and Tables

**Figure 1 animals-13-01068-f001:**
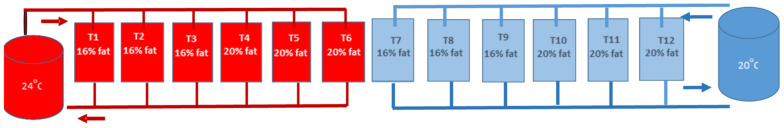
Schematic representation of the experimental set-up with D. labrax exposed to different temperatures and diets.

**Figure 2 animals-13-01068-f002:**
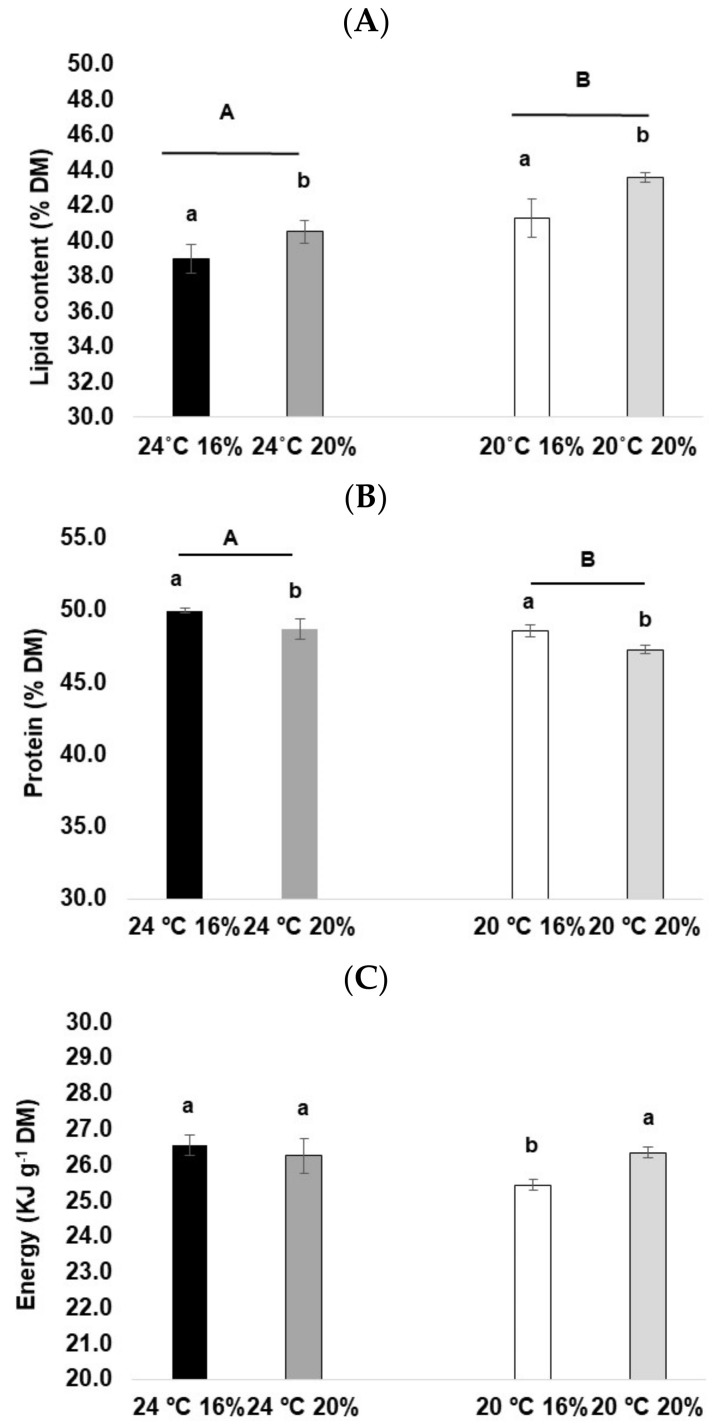
Composition analysis of *D. labrax* exposed to two different temperatures and diets. (**A**) Lipid content and (**B**) protein content, expressed in % of dry matter (DM). (**C**) Energy content, expressed in KJ g^−1^ dry matter (DM), where the values represent the mean (±SE). Lowercase letters (a,b) indicate significant differences between diets, while uppercase letters (A,B) indicate significant differences between temperature treatments.

**Figure 3 animals-13-01068-f003:**
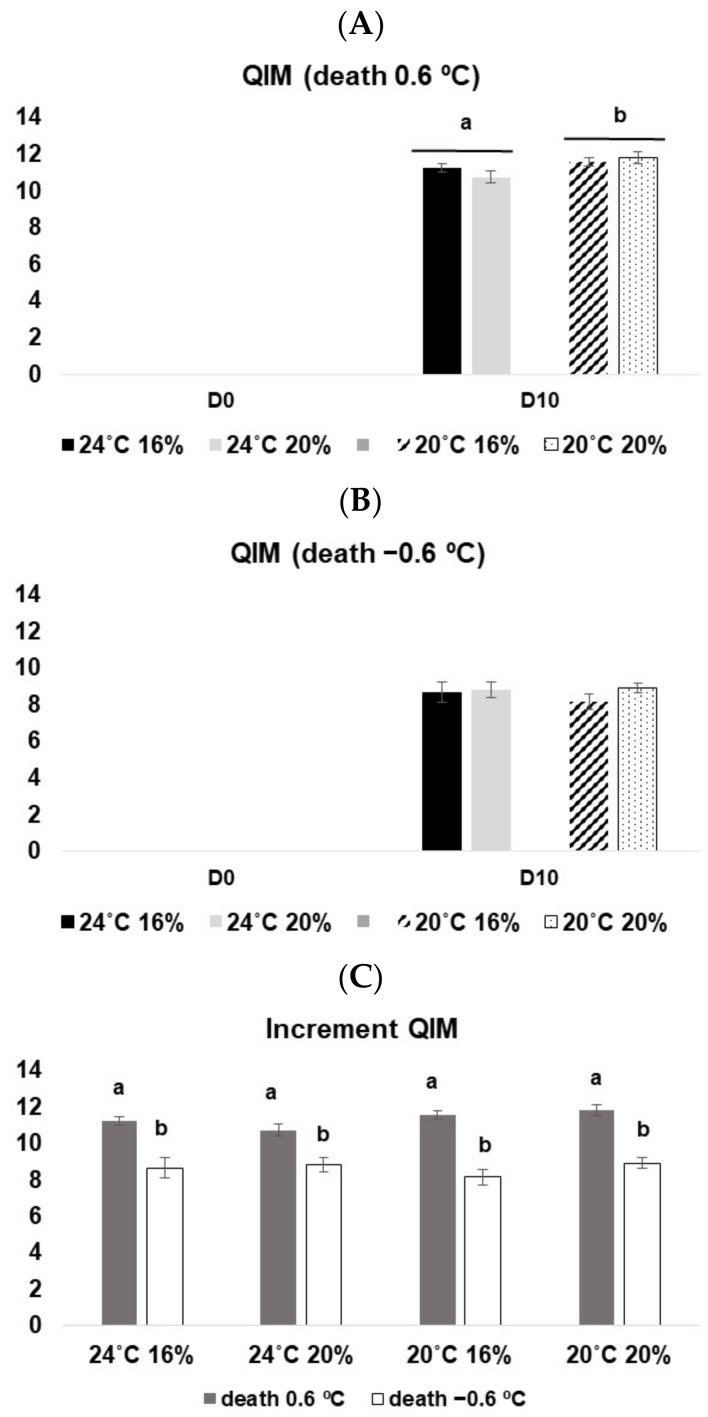
QIM values for *D. labrax* along with 10 days in ice. (**A**) fish sacrificed at 0.6 °C; (**B**) fish sacrificed at −0.6 °C; (**C**) increment of QIM from D0 to D10 at both death conditions. The values represent the mean ± SE. Different letters (a,b) represent statistical differences among treatments.

**Figure 4 animals-13-01068-f004:**
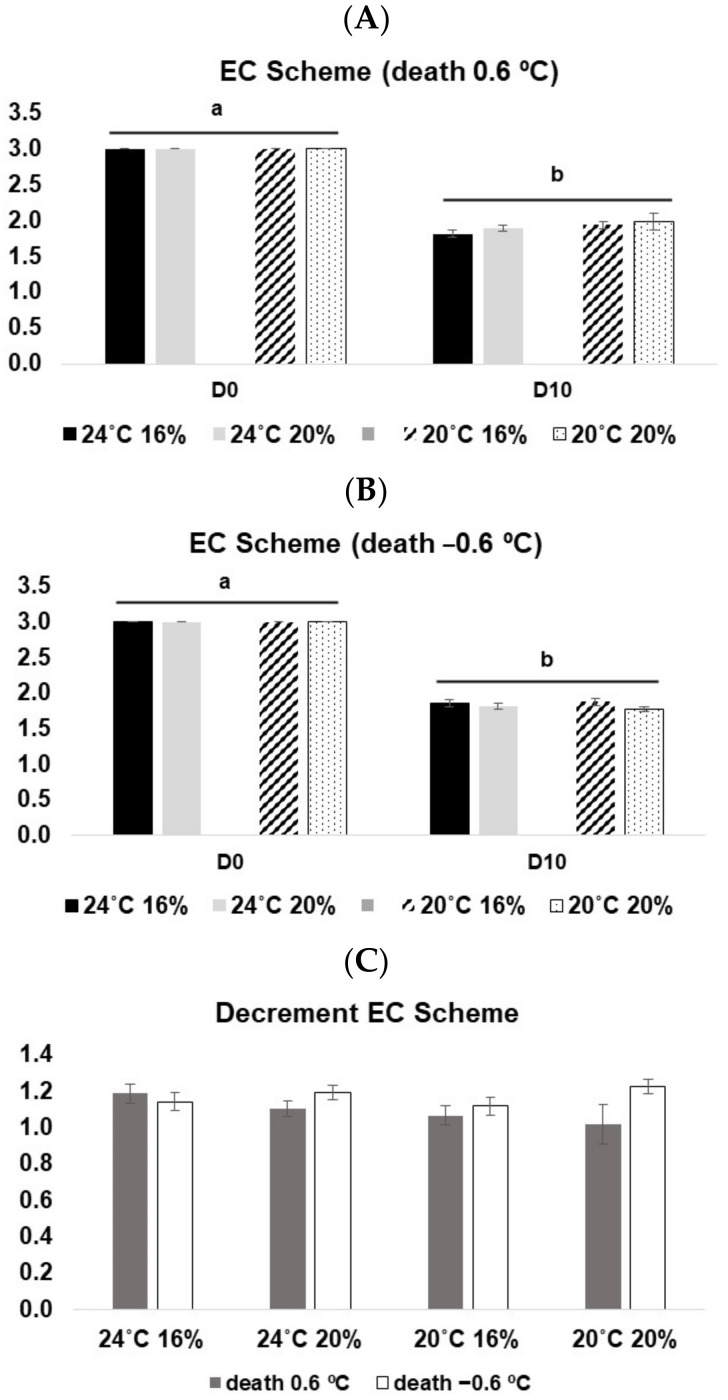
EC scheme values for *D. labrax* along with 10 days in ice. (**A**) fish sacrificed at 0.6 °C; (**B**) fish sacrificed at −0.6 °C; (**C**) decrement of the EC scheme from D0 to D10 at both death conditions. The values represent the mean ± SE. Different letters represent statistical differences among treatments.

**Figure 5 animals-13-01068-f005:**
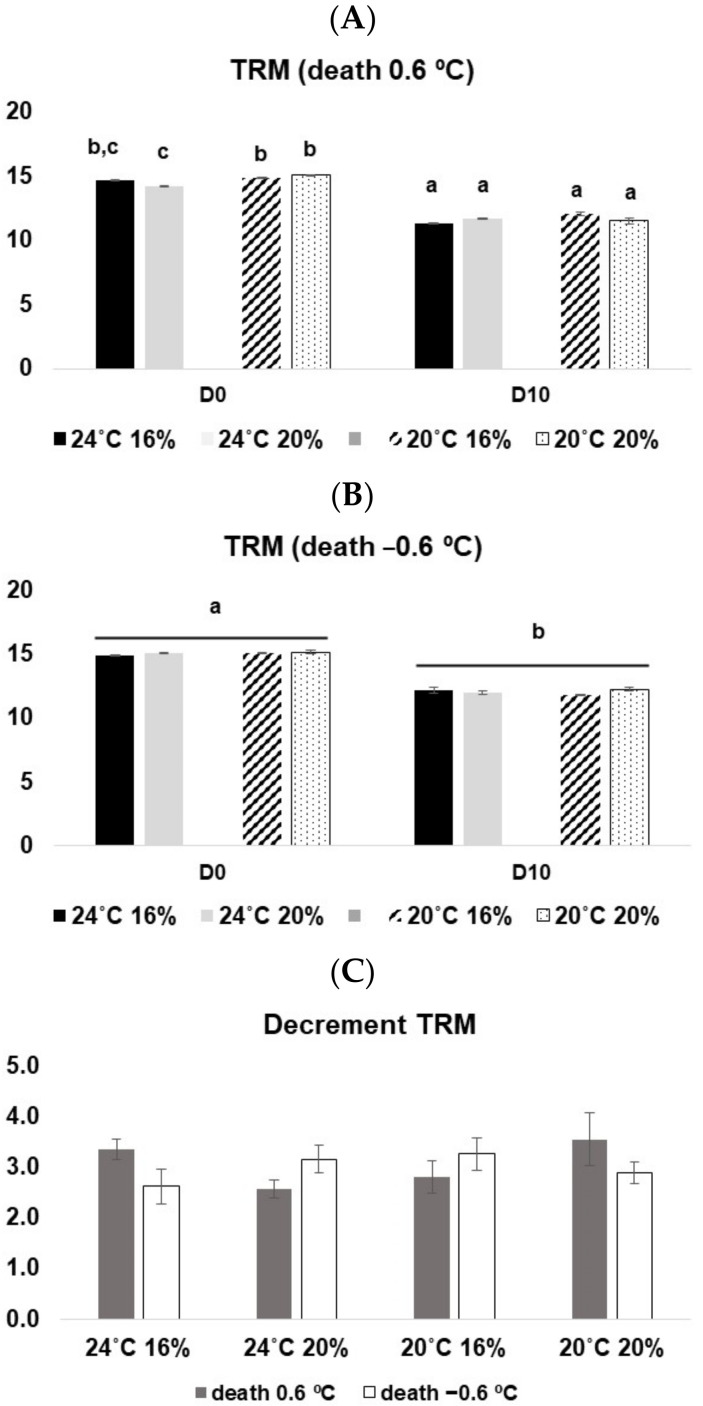
TRM values for *D. labrax* along with 10 days in ice. (**A**) fish sacrificed at 0.6 °C; (**B**) fish sacrificed at −0.6 °C; (**C**) decrement of TRM from D0 to D10 at both death conditions. The values represent the mean ± SE. Different letters represent statistical differences among treatments.

**Figure 6 animals-13-01068-f006:**
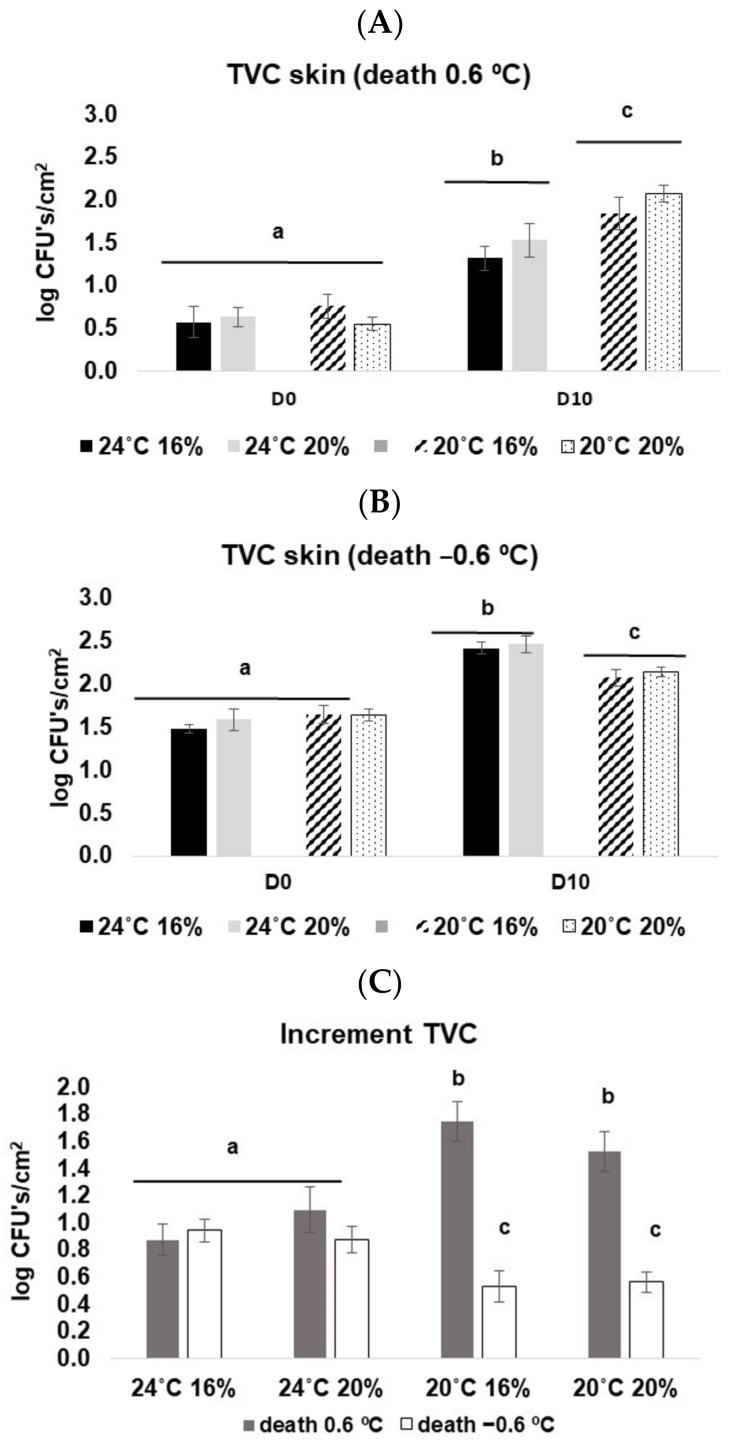
TVC values for *D. labrax* along with 10 days in ice. (**A**) fish sacrificed at 0.6 °C; (**B**) fish sacrificed at −0.6 °C; and (**C**) an increment of TVC from D0 to D10 at both death conditions. The values represent the mean ± SE. Different letters represent statistical differences among treatments.

**Figure 7 animals-13-01068-f007:**
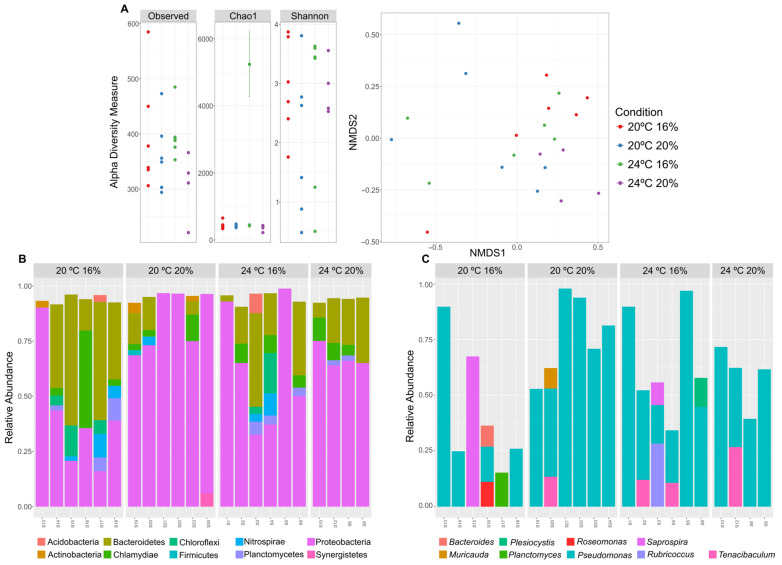
Alpha and beta-diversity metrics (**A**) and taxonomical structure of the skin microbiome at D0 of *D. labrax* sacrificed at 0.6 °C are represented as the top-10 most abundant (>2% of relative abundance) bacterial phyla (**B**) and genera (**C**) in all samples.

**Figure 8 animals-13-01068-f008:**
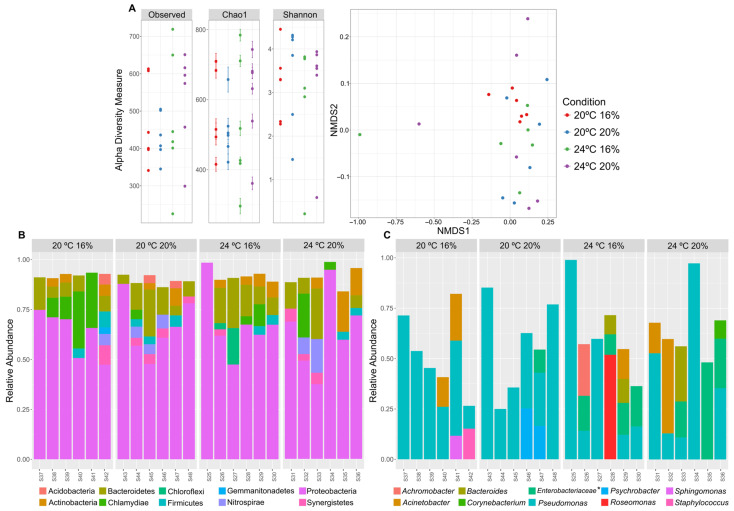
Alpha and beta-diversity metrics (**A**) and taxonomical structure of the skin microbiome at D10 of *D. labrax* sacrificed at 0.6 °C are represented as the top-10 most abundant (>2% of relative abundance) bacterial phyla (**B**) and genera (**C**) in all samples. * Unresolved genus belonging to the Enterobacteriaceae family.

**Figure 9 animals-13-01068-f009:**
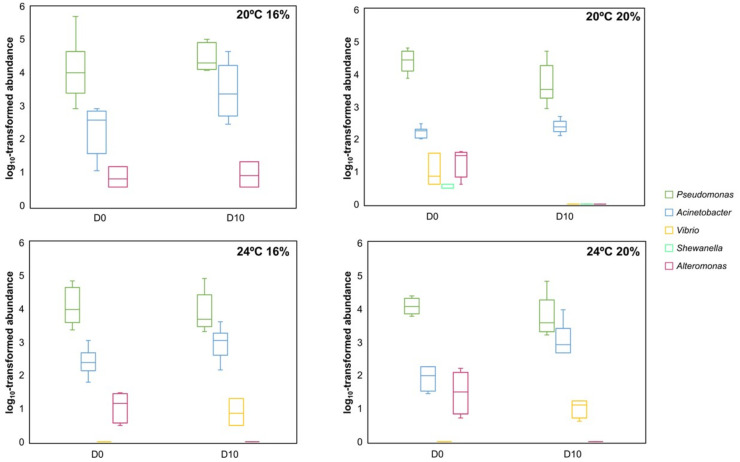
Prevalence and distribution of genera associated with fish spoilage bacteria in the different experimental conditions during the 10-day ice storage period. Boxplots show the minimum, maximum, and median log10-transformed abundances of each bacterial genus.

**Table 1 animals-13-01068-t001:** Parameters of growth evaluation and condition index of *D. labrax* exposed to different temperatures and diets. The values represent the mean ± SD. Different letters represent significant differences among treatments. Only for DGI were there significant differences between temperatures.

	24 °C 16%	24 °C 20%	20 °C 16%	20 °C 20%
**Initial Body Weight (IBW)**	60.54 ± 4.93	56.66 ± 2.64	60.55 ± 3.84	60.69 ± 2.86
**Final Body Weight (FBW)**	141.98 ± 13.77	136.71 ± 6.15	129.53 ± 13.05	133.97 ± 8.35
**Weight gain (WG)**	81.44 ± 8.91	80.05 ± 3.63	68.88 ± 10.34	73.28 ± 5.64
**Feed Conversion Ratio (FCR)**	1.36 ± 0.16	1.46 ± 0.01	1.41 ± 0.16	1.46 ± 0.11
**Daily Growth Index (DGI)**	2.30 ± 0.12 ^a^	2.34 ± 0.04 ^a^	2.01 ± 0.21 ^b^	2.12 ± 0.09 ^b^
**Protein Efficiency Ratio (PER)**	1.46 ± 0.16	1.36 ± 0.05	1.37 ± 0.29	1.4 ± 0.08
**Condition Index (CI)**	1.2 ± 0.09	1.2 ± 0.13	1.2 ± 0.13	1.2 ± 0.14

## Data Availability

The data presented in this study are available on request from the corresponding author.

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
