# Peer review of "Combined Effects of Temperature and Dietary Lipid Level on Body Composition, Growth, and Freshness Profile in European Seabass, Dicentrarchus labrax"

_animals, 2023, doi:10.3390/ani13061068_

Round 1

Reviewer 1 Report

The study covers important aspects of the Seabass, a commercial important species. However, the points as mentioned have not addressed

1. Design of the experiment particularly the impact/effect of different temperatures on the growth and freshness is not conviencing. Authors have not mentioned the selection of temperature range. Why 20 0C and 240C? Why not >24 0C or <20 0C

2. For microbial analysis, the gill tissues must have been included in addition to skin swab to check the freshness.

3. Reason for taxonomic identification of microbes on skin is not also convincing

Hence, the MS is not recommended for the publication

Author Response

1.Design of the experiment particularly the impact/effect of different temperatures on the growth and freshness is not convincing. Authors have not mentioned the selection of temperature range. Why 20 0C and 240C? Why not >24 0C or <20 0C

The range of temperatures chosen for the experimental set-up were based on the following explanations. The 20 ºC corresponds to an average summer sea surface temperature for the Portuguese coast, so it was considered as an ambient temperature. The 24 ºC corresponds to a warming scenario (+ 4 ºC). According to [22] it is estimated until the end of this century an increase of temperature of 4 ºC. This information was added to the manuscript, page 6, lines 139-141.

  1. For microbial analysis, the gill tissues must have been included in addition to skin swab to check the freshness.

We thank the reviewer’s suggestion. Regarding the microbial analysis we decided to analyse just the skin tissues since, first, according to the literature the bacterial counts on the skin and gills usually yield proportional results [28,29]. Fresh fish usually present skin bacterial counts between 102-103, while gills present very similar numbers (between 102-104), and it is known that these numbers increase proportionately as freshness is lost. So, microbial analysis of skin will give us a similar picture in terms of freshness profile as gills microbial counts, in spite of the latter possibly resulting in slightly higher counts. In addition, we consider that the skin is more relevant to the consumer since in some cases the gills are removed before the fish reaching the consumer. So, in terms of cost-benefit, and attending to some human resources limitations we considered that it would be more beneficial to study the microbiology of the skin.

We added this information in the materials and methods, section 2.6.3. Please check page 10, lines 222-229.

  1. Reason for taxonomic identification of microbes on skin is not also convincing

We thank the reviewer’s comment. We analysed the taxonomic profiling of fish skin microbiome by 16S rRNA gene metabarcoding in order to help elucidating, through culture-independent methods, how bacterial taxa in general and also spoilage-associated taxa evolved over the time of storage of fish on ice, information that, in our opinion, is very relevant and enriched our study. To clarify the reason that led us to perform this analysis, we added this information to the Discussion section (page 18, line 429-433).

Reviewer 2 Report

This paper investigated effects of water temperature and dietary lipid on European sea bass, which has significant scientific value. However, this paper need improvement in area of abstract, data presentation and interpretation. Some of the parameters investigated, however, not property discussed. Please find specific comments in attached documents. 

Author Response

According to referee 2, we did the following modifications:

In general the English was improved along the text and more relevant references were added to the document.

1.As suggested by the referee, the simple summary suffered modifications in agreement. The expression “rearing conditions” was not conveniently applied, so we modified the sentences in the simple summary where it appeared. Please check lines 37-39 and 45-47.

  1. In the abstract, all the suggested modifications were done. Regarding the last comment, where the referee asks if there is any relation of killing methods with dietary lipid and rearing temperature, we would like to clarify that there is no relationship between them. The main objective of the paper was to evaluate the effects of temperature and diet lipids on the growth and freshness of European seabass. Besides this, we decided to evaluate the effect of two different killing temperatures on the freshness profile. So, in fact there is no relation between the experimental factors and the killing methods.

  1. In the introduction, all the suggested modifications were done. In the last comment of the introduction, the word “rearing” was deleted since it was not appropriated.

  1. In the materials and methods, section 2.1, it was indicated the number of fishes per tank (i.e. 17 individuals/tank). Please check page 6, lines 137-138.

  1. In the materials and methods, page 7, section 2.3, it was indicated in lines 157-158 how the temperature of the tanks for the two killing methods were maintained. Basically, the two temperatures were controlled with a thermometer and whenever necessary more ice was added to the water.

  1. In the materials and methods, page 7, section 2.4, line 169, it was corrected the time used for the analysis of dry matter. In spite of 1h should be 24h. “was analysed for dry matter (AOAC 2003.05, 105 °C, 24 h)”.

  1. As suggested by the referee, in table 1 we indicated the statistical differences with lettering. In this case we just obtained significant differences for the daily growth index (DGI). For the FBW there were no statistical differences among treatments.

  1. In section 3.2.3, the TRM analysis was not related to the treatments.

  1. In the discussion, second paragraph, lines 383-384, the sentence was corrected. “These results seem to be in accordance with [35,36] that concluded that higher temperature resulted in lower concentrations of lipids.”

  1. In the discussion, third paragraph was completely modified. “On the other hand, the protein content declined in fish reared at 20 oC and fed with 20% lipid diet. Regarding the effect of temperature, our findings are in accordance with [38], for the Asian seabass, that observed that body protein was lower at lower temperatures. However, a contrary response was observed for the pikeperch Sander lucioperca (i.e. higher protein content when exposed to lower temperature) [39]. In addition, concerning the effects of diet, our results are in agreement with [40] that observed lower percentage of crude body protein in fishes exposed to higher dietary lipid.” Please check page 16, lines 387-394.

  1. In the discussion, page 17, paragraph 2 was modified in accordance with the referee comments. Please check lines 401-407.

“In addition, our results also demonstrated that the increase of temperature from 20 ºC to 24 ºC had a positive effect on the growth performance. Fish reared at 24 ºC had a significantly higher daily growth index than those exposed to 20 ºC. This pattern was previously observed in other studies, like the one of [46], in which specific growth rate increased with temperature up to a maximum at 25 ºC. According to the literature, growth rate increases with increasing temperature, however for extreme temperatures the effect can be negative [35].”

  1. As suggested by the referee, the results of the metabarcoding analyses were added to the abstract. Please check the new lines 60-63.

  1. The scientific name D. labrax was replaced along the text by European seabass.

  1. In the discussion, page 19, paragraph 2 was totally modified. “It is also curious that the number of TVC at D0 was three times higher in fish sacrificed at -0.6 ºC compared with fish sacrificed at 0.6 ºC. This can be related with the fact that fish sacrificed at -0.6 ºC produced more mucus than at 0.6 ºC (visual observations), which can promote the development of a bacterial community. This mucus production can be related with stressful environmental conditions (e.g. salinity, oxygen, nutrients and temperature changes) for the fish, in this case the decline of temperature. According to previous studies, changes in skin mucus production and composition are a response of the fish to cope with changes in the surrounding water [44]. Effectively, many authors suggested that the number of skin mucous cells can be used as a stress index for fish [45].” Please check lines 443-451.

Reviewer 3 Report

The article "Combined effects of temperature and dietary lipid level on body composition, growth and freshness profile in European seabass, Dicentrarchus labrax" presented for review concerns a very interesting aspect of the influence of variable rearing conditions, diet and pre- and post-slaughter conditions on the meat quality of the tested fish species .

The introduction to the study clearly explains the intentions of the authors who, setting the goal and pointing to the novelty, appropriately designed, conducted and described the experiment and its results.

While reading the manuscript, several inaccuracies were found:

1. Vas-Pires and Ozorio autocitations found 4 times, please limit their number.

2. Please correct paragraph 2.4 for the font size.

3. Figures 1 to 8 are hard to read. Letters and numbers are blurred, which makes it difficult to read the study. Please improve the sharpness of the fonts.

4. In studies on stored products, especially fish, together with the activity of microorganisms, there are also products of protein breakdown, including biogenic amines, such as histamine, which is a criterion for food safety. Such results would significantly increase the value of the study. Perhaps it would be worth adding these results (if such tests were performed)?

Author Response

According to referee 3, we did the following modifications:

1.Vas-Pires and Ozorio autocitations found 4 times, please limit their number.

As suggested by the referee we replaced some of the citations of Vaz-Pires and Ozório by others. For example, Agueria et al 2016 was replaced by Estevez and Anibal 2021. Freitas et al. 2019 was replaced by Prabhakar et al 2020 and Ozório et al 2006 was replaced by Katersky et al 2007.

2.Please correct paragraph 2.4 for the font size.

We do not understand why the font size has changed, since in our document is ok along the manuscript.

  1. Figures 1 to 8 are hard to read. Letters and numbers are blurred, which makes it difficult to read the study. Please improve the sharpness of the fonts.

The size and type of font were modified to be more visible.

4.In studies on stored products, especially fish, together with the activity of microorganisms, there are also products of protein breakdown, including biogenic amines, such as histamine, which is a criterion for food safety. Such results would significantly increase the value of the study. Perhaps it would be worth adding these results (if such tests were performed)?

We agree with the referee, but these analyses were not performed at the moment of the experiment, so we cannot add to the manuscript.

Reviewer 4 Report

Dear Author,

The work is written in clear and understandable language, the research and descriptive content is presented concretely and to the point. I present minor reservations and comments in text. 

Sincerely,

Author Response

According to referee 4, we improved the English along the manuscript and answered to the following questions:

1.As suggested by the referee, we added in the introduction, page 5, information regarding the biology of the European seabass. European seabass, Dicentrarchus labrax, is a coastal marine teleost that lives in shallow waters (< 100 m) with a wide geographical distribution through the coasts of the Mediterranean Sea, the Black Sea, and the eastern Atlantic Ocean from Norway to Morocco, the Canary Islands, and Senegal [14]. It is also extensively farmed at different conditions (e.g. open and semi-open culture system), so it is able to cope with daily and seasonally oscillations in environmental conditions, namely sea surface temperature [15]. It can reach a maximum length of 1 m but rarely exceeds 50 cm [16]. European seabass is carnivorous, feeding plankton, fish and crustaceans [17]. Please check lines 109-116.

2.In material and methods, section 2.1, we have included the average size of the seabass (16.8 ± 0.6 cm), as requested by the referee. Please check page 6, line 136.

3.As suggested by the referee, the salinity units were introduced in the text (i.e 35.5 ± 0.7 ppt). Please check page 6, line 144.

4.In the discussion, page 16, paragraph 4 was modified as follow:

«On the other hand, the protein content declined in fish reared at 20 oC and fed with 20% lipid diet. Regarding the effect of temperature, our findings are in accordance with [41], for the Asian seabass, that observed that body protein was lower at lower temperatures. However, a contrary response was observed for the pikeperch Sander lucioperca (i.e. higher protein content when exposed to lower temperature) [42]. In addition, concerning the effects of diet, our results are in agreement with [43] that observed lower percentage of crude body protein in fishes exposed to higher dietary lipid.

The results obtained were compared to other works, in which we can conclude that the protein content of the fish can be variable in response to temperature. So, in some works the protein content increased with exposure to lower temperature (Wang et al. 2009), while in others there were no differences in protein content between different temperature treaments (Han et al. 2003). In the present study the protein content declined when the fishes were exposed to lower temperature. So, the response can be quite variable depending, for example on the species, size, etc.» Please check page 16, lines 386-393.

5.Page 18, last paragraph, regarding the effects of temperature on growth performance, the text was modified as suggested by the referee, as follows:

In addition, our results also demonstrated that the increase of temperature from 20 ºC to 24 ºC had a positive effect on the growth performance. Fish reared at 24 ºC had a significantly higher daily growth index than those exposed to 20 ºC. This pattern was previously observed in other studies, like the one of [46], in which specific growth rate increased with temperature up to a maximum at 25 ºC. According to the literature, growth rate increases with increasing temperature, however for extreme temperatures the effect can be negative [35]. Please check page 17, lines 401-407.